# A Critical Review of the Role of the Cannabinoid Compounds Δ^9^-Tetrahydrocannabinol (Δ^9^-THC) and Cannabidiol (CBD) and their Combination in Multiple Sclerosis Treatment

**DOI:** 10.3390/molecules25214930

**Published:** 2020-10-25

**Authors:** Éamon Jones, Styliani Vlachou

**Affiliations:** Behavioural Neuroscience Laboratory, Neuropsychopharmacology Division, School of Psychology, Faculty of Science and Health, Dublin City University, Glasnevin, Dublin 9, Ireland; eamon.jones5@mail.dcu.ie

**Keywords:** multiple sclerosis, cannabinoid, Δ^9^-tetrahydrocannabinol, cannabidiol, spasticity, neuropathic pain, inflammation, experimental autoimmune encephalomyelitis, neuroprotection, cognition, animal models

## Abstract

Many people with MS (pwMS) use unregulated cannabis or cannabis products to treat the symptoms associated with the disease. In line with this, Sativex, a synthetic combination of cannabidiol (CBD) and Δ^9^-tetrahydrocannabinol (Δ^9^-THC) has been approved to treat symptoms of spasticity. In animals, CBD is effective in reducing the amounts of T-cell infiltrates in the spinal cord, suggesting CBD has anti-inflammatory properties. By doing this, CBD has shown to delay symptom onset in animal models of multiple sclerosis and slow disease progression. Importantly, combinations of CBD and Δ^9^-THC appear more effective in treating animal models of multiple sclerosis. While CBD reduces the amounts of cell infiltrates in the spinal cord, Δ^9^-THC reduces scores of spasticity. In human studies, the results are less encouraging and conflict with the findings in animals. Drugs which deliver a combination of Δ^9^-THC and CBD in a 1:1 ratio appear to be only moderately effective in reducing spasticity scores, but appear to be almost as effective as current front-line treatments and cause less severe side effects than other treatments, such as baclofen (a GABA-B receptor agonist) and tizanidine (an α2 adrenergic receptor agonist). The findings of the studies reviewed suggest that cannabinoids may help treat neuropathic pain in pwMS as an add-on therapy to already established pain treatments. It is important to note that treatment with cannabinoid compounds may cause significant cognitive dysfunction. Long term double-blind placebo studies are greatly needed to further our understanding of the role of cannabinoids in multiple sclerosis treatment.

## 1. Introduction

The earliest use of the cannabis plant can be dated back to 400 B.C. in China, where it was cultivated for the production of string, rope, and paper. The first documented use of cannabis as a medicine was not until the first century of the common era, in the oldest known pharmacopoeia, that of Pen-Ts’ao Ching [1]. For centuries since, the cannabis plant has been used both for recreational and for medicinal purposes, despite the biological and scientific underpinning of its mechanism of action being unknown until a few decades ago. In the 1920s, a smear campaign was run in America which claimed that cannabis turned anyone who indulged in it into drug-crazed violent criminals. This led to a rapid introduction of restrictive laws, some of which are still in place, and cannabis was classified as a narcotic, which carries opioid undertones, furthering the stigma associated with cannabis use [2].

Currently approximately 124 million people use cannabis annually, with the most popular method of administration being smoking and inhaling the flower in a joint [3]. Despite its potential for both physical and psychological dependence being relatively low [4,5,6], cannabis is a class C drug in the UK and Ireland and has varying degrees of legality in the United States and further afield. Despite being legalized in some countries and regions, cannabis use is still marginalized and considered a vice [7]. A shift in this attitude can be seen between 1980 and 1990 where cannabis started being prescribed as an anti-emetic medication and an appetite increaser in HIV and cancer patients, as well as for the reduction of intraocular pressure in glaucoma patients [5,6].

For the medical use of cannabis in multiple sclerosis (MS), there are four main routes of administration, these being: oral administration, mucosal administration, subcutaneous administration (SC), and transdermal administration [8]. There is evidence to suggest that up to 4% of pwMS currently have access to, or have previously illegally accessed, unregulated cannabis products to treat their disease symptomatology [6,7].

### 1.1. Multiple Sclerosis, Neuropathology, Treatment Avenues and Outcomes

MS is a degenerative neuroinflammatory disease affecting the central nervous system (CNS). It is characterised by significant demyelinating plaques in both grey and white matter that can be identified by the loss of myelin and the associated oligodendrocytes presumably of an autoimmune aetiology [8,9]. The degree of permanent clinical deficits observed is more closely related to the degree of axonal atrophy and loss compared to the extent of demyelination or the number of lesions present [10]. These inflammatory lesions are composed mainly of T-lymphocytes and lesser amounts of B-lymphocytes, plasma, microglia, and macrophages. These types of cells seem to be actively involved in the demyelinating process. The presence of histocompatibility antigens, cell adhesion molecules and cytokines expressed in the lesions also suggests a T-cell mediated immuno-response is responsible for the neuroinflammation in MS [11].

MS can be thought of as a disorder with several subtypes and variants. Three main types of MS have been identified; (1) relapsing-remitting MS, which consists of episodes of worsening or new symptoms. (2) primary progressive MS, which consists of a linear worsening of symptoms over several years, but there are still periods where symptoms improve and (3) secondary progressive MS, where symptoms worsen and there are no longer signs of remission. The presentation of symptoms and the symptom severity depend on where the demyelinating plaques are located in the CNS [12].

As is commonly known, there is currently no cure for MS. Despite this, there are two main pharmacotherapeutic avenues: the first of them includes disease-altering drugs which can modify the course of the disease, such as recombinant interferon β-1a (e.g., Avonex), recombinant interferon β-1b (e.g., Betaferon) and glatiramer acetate [13], which appear to not only help prevent relapse but also to improve neuropsychological outcomes in pwMS [14]. They do this by blocking gamma interferon and inducing the release of anti-inflammatory cells [15]. The second pharmacotherapeutic avenue includes symptom-altering drugs which can improve the patient’s quality of life by improving pain and spasticity and mood issues associated with MS, such as the GABA-B receptor agonist baclofen and the α2 adrenergic receptor agonist tizanidine [16].

### 1.2. The Endocannabinoid System and Multiple Sclerosis

For centuries, both recreational and medicinal use of cannabis was blind. The biological explanations for its effects were unknown. With a spike in recreational drug use in America during the 1960s, a similar spike was seen in studying its effects [17]. From studying the endocannabinoid system, researchers have been able to understand the impact cannabis compounds can have on the body and mind [13,14,15].

The endocannabinoid system is comprised of at least two types of Class A G-protein-coupled receptors which exhibit primarily Gi/o signaling mechanisms: the cannabinoid 1 receptor (CB_1_) and the cannabinoid 2 receptor (CB_2_) which can be activated by endogenous cannabinoids, synthesized on demand, or exogenously administered compounds [18]. Receptors for these endogenous ligands can be found in the CNS, both in the brain and spinal cord, mainly in the frontal cortex, basal ganglia and cerebellum, and diffusely around the body, most notably in the gastrointestinal tract, adipose tissue, liver and connective tissues [19]. Importantly, cannabinoid receptors can be found in the presynaptic junction, thus engaging in retrograde signaling, a specific feature of this biochemical system [20].

The first endogenous cannabinoid ligand identified, *N*-arachidonoylethanolamide, shortened to anandamide after the Sanskrit word Ananda which means “bliss”, was first discovered in 1992 by mass spectrometry and magnetic resonance spectrometry [21]. Following shortly after, the second endogenous cannabinoid ligand 2-arachidonoylglycerol (2-AG) was discovered [22]. These endo-cannabinoid compounds are produced immediately before excitatory release after being synthesised from fatty compounds in the cell membrane. Once these compounds have been released mostly from the cell body and dendrites, they exert their effect on presynaptic neurons [17]. Interestingly, the most recently defined expanded endocannabinoid system, i.e., the endocannabinoidome, includes several mediators that are biochemically related to the endocannabinoids, and their receptors and metabolic enzymes [23], all of which seem to play a role in the therapeutic effects on MS and other neurodegenerative conditions.

Cannabinoids are complex and include constituents in the classes of polyketides, terpenoids, sugars, alkaloids, flavonoids, and quinones; they are a class of meroterpenoids. The cannabis plant has two compounds that have been studied extensively, and both are found in greatest abundance in a waxy resin surrounding the cannabis plant leaves and flower [24] The first one is Δ^9^-tetrahydrocannabinol (Δ^9^-THC), the main psychoactive constituent of the cannabis plant and the most abundant one, which was first discovered in 1964 by Gaoni and Mechoulam [25], and confirmed by synthesis and X-ray crystallography as a benzenesulfonate ester. The second component is cannabidiol (CBD), a non-psychoactive cannabinoid ligand, whose chemical structure was successfully isolated in 1940 and elucidated by Mechoulam and Shvo in 1963 [26]; other pharmacologically interesting compounds are cannabinol (CBN), cannabinoid acids and cannabivarin (CBV).

In regards to the chemical structure and biosynthetic pathways of cannabinoids, including Δ^9^-THC and CBD, there is a structural similarity comprising a resorcinol (A-ring) and terpinoid moiety (C-ring). Terpenoids are formulated starting from the (non)-mevalonate pathways which produce dimethylallyl pyrophosphate (DMAPP) and isopentenyl pyrophosphate (IPP). Both DMAPP and IPP are then coupled by geranyl pyrophosphate synthase to form geranyl pyrophosphate (GPP) which is a precursor for many terpenoids [27]. Further, coupling of olivetolic acid with GPP by geranyl transferase (i.e., through the assembly and modification of a C12 polyketide unit and a monoterpene unit from the deoxyxylulose phosphate pathway) forms cannabigerolic acid (CBGA), which is the biosynthetic starting point for most cannabinoids and a direct precursor of CBD and Δ^9^-THC [28]. CBD is the result of CBGA oxidative cyclisation which results in the formation of a link between C-1 and C-6 of the prenyl unit [29]. Importantly, very recent findings show that a significant amount of CBD transforms to Δ^9^-THC in a hot gas chromatography injection system when acidic precipitation agents, such as TFA, TCA, HClO_4_, H_2_SO_4_, ZnSO_4_ or CHCl_3_, are used for plasma protein precipitation, a finding that can be very important not only for the definition of effective doses and plasma concentrations by pharmacologists and medical doctors examining the pharmacokinetics of CBD-containing drugs for therapeutic purposes, but also for forensic scientists who may find innocent people guilty of using marijuana or its preparations by mistake [30].

Δ^9^-THC, a highly lipophilic (i.e., low water soluble) low vapor pressure resinous oil with a light yellow colour, is a 21-carbon terpenoid with an internal double bond (C9−C10) and two stereocenters (6a, 10a), with the levorotatory trans stereoisomer, (−)-*trans*-Δ^9^-THC (6aR, 10aR) [31]. Δ^9^-THC, Δ^9^-THC acid (Δ^9^-THCA) and cannabidiolic acid (CBDA) are the results of CBGA cyclisation through a cationic intermediate with positive charge at C-3 [32]. Specifically for the synthesis of CBDA and Δ^9^-THCA, a stereoselective ring-closure induced either by CBDA or THCA synthase needs to take place, while CBN and CBD subvariants are also formulated from CBGA. [27,33]. Please see Figure 1 below for the structure of both Δ^9^-THC and CBD.

Δ^9^-THC activates both CB_1_ and CB_2_. It has the highest potency at both CB_1_ and CB_2_ compared to all other phytocannabinoids. However, it acts as a moderate partial agonist towards both receptors because it cannot induce their full activation to produce a maximal response. Thus, as a partial agonist, it presents a mixed agonist-antagonist profile depending on the cell type, expression of receptors and presence of endocannabinoids or other full agonists [34]. When compared to the endogenous cannabinoids, the CB_1_ affinity and potency of Δ^9^-THC is comparable to that of both AEA and 2AG. Δ^9^-THC also resembles AEA in its functional properties (i.e., they are both partial agonists) and differs from 2AG, which induces a full response from both receptors as a full agonist [35].

Except for its effects at CB_1_ and CB_2,_ Δ^9^-THC acts as a potent agonist for the putative cannabinoid receptor GPR18, while it also exhibits activation of GPR55 in different assays, without affecting ERK1/2 phosphorylation or β-arrestin recruitment, a discrepancy that needs to be studied further. Moreover, Δ^9^-THC does not affect the vanilloid type 1 receptor (TRPV1; i.e., the capsaicin receptor), while it shows agonistic effects at the TRPV2, TRPV3, and TRPV4 channels [34]. Δ^9^-THC also appears to be a serotonin 5HT_3A_ receptor antagonist and an allosteric modulator of the opioid receptors. Specific non-G-protein-coupled receptors have also been suggested as agonistic targets of Δ^9^-THC, such the peroxisome proliferator activated receptors (PPARs), specifically of the PPAR-gamma subtype, a non-cannabinoid receptor situated on the cell’s nucleus. PPARs are a group of three nuclear receptors, namely PPAR-α, PPAR-γ, and PPAR-δ. PPARs are triggered by hormones, endogenous fatty acids, and various nutritional compounds. When activated, PPARs bind to certain segments of DNA to promote or prevent transcription of specific genes. Many of the genes regulated by PPARs are involved in energy homeostasis, lipid uptake and metabolism, insulin sensitivity, and other metabolic functions. Δ^9^-THC antitumor effects, as well as those of vascular relaxation have been linked to its agonism at PPAR-gamma. Further, low concentrations of Δ^9^-THC significantly enhance the amplitudes of glycine-activated currents and it is this activity of Δ^9^-THC at the glycine receptors (GlyRs) that seems to contribute to the cannabis-induced analgesia in mouse models [34].

Δ^9^-THC undergoes hepatic metabolism to 11-hydroxy-Δ^9^-THC, which is more potent than Δ^9^-THC and also crosses the blood barrier. Δ^9^-THC has a short plasma half-life, because it rapidly diffuses into lipid tissue, but the biological half-life is prolonged. Thus, the tissue elimination half-life of Δ^9^-THC is about 7 days, and complete elimination of a single dose may take up to 30 days.

The chemistry, pharmacology and molecular targets of CBD have recently been reviewed extensively [36]. Thus, briefly here, unlike Δ^9^-THC, the more neglected for decades phytocannabinoid CBD has little binding affinity for either CB_1_ or CB_2_, which likely accounts for its lack of psychotropic activity. Interestingly, and similarly to Δ^9^-THC, CBD is also considered a multi-target compound. At low micromolar to sub-micromolar concentrations, CBD inhibits the equilibrative nucleoside transporter (ENT), the orphan G-protein-coupled receptor GPR55, and the transient receptor potential of melastatin type 8 (TRPM8) channel [37]. On the other hand, CBD enhances the activity of the 5-HT_1a_ receptor, the a3 and a1 GlyRs, the transient receptor potential of ankyrin type 1 (TRPA1) channel, and has a bidirectional effect on intracellular calcium. Quantitative results of CBD’s membrane protein interactions by target, cell type, and IC_50_ or EC_50_ indicate that CBD IC_50_s/EC_50_s are considerably similar between all its targets (with the exception of GlyRs), with IC_50_s ranging between 0.06–4 µM for the different receptor targets [38], while for the sodium channels specifically the IC_50_ range is 1.9–3.8 μM [39].

At higher micromolar concentrations, CBD acts primarily through receptor-independent channels and by binding with various non-cannabinoid receptors. It activates the TRPV1 and TRPV2 channels (EC_50_ = 3.2 ± 3.5 μΜ) [40]. Further, recent studies indicate that CBD influences the expression of some genes by directly activating PPARs, specifically the PPAR-gamma subtype. It also promotes PPAR-α activity by inhibiting fatty acid amide hydrolase (FAAH). The ability of CBD to activate PPAR-γ has promising therapeutic implications, particularly with respect to cancer and metabolic disorders [41]. For a very recent and detailed presentation of phytocannabinoid concentrations and corresponding receptor potencies of cannabis extracts, as well as pure phytocannabinoids and reference compounds, please see the very interesting study by Yang and colleagues [42]. At a very recent systematic review on the pharmacokinetics of CBD in humans [43], it was found that, while in animal studies oral bioavailability is low, in human studies, the half-life of CBD was reported between 1.4–10.9 h after oromucosal spray, 2–5 days after chronic oral administration, 24 h after intravenous administration, and 31 h after smoking.

As with other disorders, the question that arises is whether cannabinoid compounds can be used to treat MS. If so, are they effective and what is their mechanism of action in improving symptomatology? Do they help by reducing symptomatology and improving the patient’s quality of life, or are they found to be neuroprotective, thus altering the course of the disease? In order to try and identify some possible answers to these questions and identify the progress-to-date as well as the research priorities ahead, a critical analysis of the literature is presented in this review.

## 2. Animal Studies

The human immune system and CNS are possibly two of the most complex systems in the body, thus, replicating a human immune system-mediated disease that affects the CNS constitutes an exceedingly difficult task indeed. A need for animal models of MS is sustained by several limitations of human studies, such as having limited access to human MS tissue as autopsies in this disease are biased towards a late stage of disease, making studying the disease even more difficult. A further limitation is the fact that experimental drugs cannot be directly tested in humans for ethical reasons, at least not before efficacy and safety data in another species are made available. Another advantage of the development of MS animal models is the fact that experimental circumstances can be modified in animal studies in an easier, more controlled manner when compared to human studies.

By accurately imitating MS in an animal model, we can observe the efficacy of cannabinoid compounds in disease treatment. By examining the literature, three induced animal models of MS have been identified: virally induced demyelination, toxin-induced demyelination and experimental autoimmune encephalomyelitis, all with their advantages and disadvantages in terms of examining the effects of cannabinoids [28,29].

### 2.1. Animal Models of Multiple Sclerosis

#### 2.1.1. Virally-Induced Demyelination

Theiler’s Murine Encephalomyelitis Virus (TMEV) is used to induce MS, epileptic seizures and virus-induced myocarditis in susceptible mouse strains such as the SJL mice [44,45] initially reported by Max Theiler in 1934 [46]. Intracerebral infection of TMEV causes flaccid paralysis of the hind limbs and acute polioencephalomyelitis and chronic demyelination two weeks post-infection. TMEV can be found primarily in the oligodendrocytes, microglia and astrocytes with apoptotic oligodendrocytes being identified pre-chronic demyelination in the spinal cord white matter. The host immune system tends to clear the virus quickly from the brain, but not from the spinal cord [47].

Several advantages of using TMEV as a model of MS have been reported in the literature, and they include: a chronic disease that lasts the lifetime of the animal; pathology is limited to the CNS with brain, brainstem and spinal lesions in transgenic mice [48], which makes it useful for the study of specific pharmacological compounds like cannabinoids, as they also have effects outside of the CNS; T1 (longitudinal relaxation time) hypointensities in the cerebrum and T2 (transverse relaxation time) hypointensities in the thalamus similar to the human disease found by magnetic resonance imaging [27,28]; haemorrhagic demyelination corresponding to brain and spinal cord atrophy similar to those observed in the human disease [49]; and, several viral strains similar to TMEV have been identified to cause demyelinating diseases like MS in humans [47].

#### 2.1.2. Toxin-Induced Demyelination

Toxin-induced demyelination is different to virally induced demyelination in that it does not attempt to imitate MS, but mimics the processes of demyelination and remyelination of cells. This model is induced in animals by either injecting the toxin lysolecithin, a phospholipase A_2_ activator or by adding the toxin cuprizone, a copper ion chelator, to the animal’s food [27,32]. Similarly to what can be found in the plaques of humans with MS, lesion sites of animals injected with lysolecithin are infiltrated with T cells, B cells and macrophages followed by spontaneous remyelination [47]. This model can be advantageous in inducing a chronic demyelinating disease, as it is characterized by degeneration of oligodendrocytes rather than by a direct attack on the myelin sheath [50].

Building from these earlier approaches, a new model has recently been suggested; the cuprizone autoimmune encephalomyelitis (CAE) model. This model resembles the experimental autoimmune encephalomyelitis (EAE) model (described below) in both its histopathology and how it is developed. The CAE model is induced by feeding cuprizone to mice during a two-week period, followed by an injection of complete Freund’s adjuvant and pertussis toxin. After an incubation period, significant demyelination can be observed along with clear radiological abnormalities in the corpus callosum [42,43]. This creates a model that can be used to observe demyelination and remyelination progression at different time points and with gradual effects in different brain areas, and specific manipulations have recently been suggested for the development of this model depending on the exact aims and focus of each study using it [51]. Taking into account the deregulation of the endocannabinoid system in MS and its important role in neuroprotection against demyelination, this model can prove very useful for testing the effects of cannabinoid compounds on myelin repair following a demyelinating insult [52,53].

#### 2.1.3. Experimental Autoimmune Encephalomyelitis (EAE)

In the 1930s, Rivers et al. [52] discovered a model of autoimmune encephalomyelitis in mice while studying the neurological complications associated with the anti-rabies vaccine. Several years later, Wolf and colleagues noted the similarities between the human demyelinating disease and this novel EAE [54].

EAE can be induced in susceptible animals by SC injecting either myelin oligodendrocyte glycoprotein (MOG) (35–55), myelin basic protein or proteolipid protein. Once immunisation has occurred, activation and expansion of peripheral antigen-specific T-cells occurs which then enter the CNS, encounter the myelin antigen and induce the disease [47].

An important criticism of the model which should be noted is that therapeutic approaches have appeared to be successful in an EAE model of MS, but, when tested in the human disease have proved to be ineffective [55]. Despite this important criticism, EAE is by far the most used animal model of MS with, at the time of writing this manuscript, over 13 thousand published papers.

### 2.2. Cannabidiol Slows Symptom Onset in EAE Mice

CBD has been shown by several authors to reduce the severity of EAE in mice. The main questions to be asked are: what seems to be the clinically effective dose needed to see an improvement; what molecular changes occur after treatment; and most importantly, is CBD efficient in reducing the neuroinflammation associated with an animal model of MS?

A conflicting aspect of the literature are the CBD doses administered to the animals. Some researchers found that administering lower doses of CBD, between 5–20 mg/kg [25,26] is effective in reducing signs of the disease whereas other researchers found that a much higher dose of 50 mg/kg is the dose that most greatly improved results of scoring methods and preserved some paw sensitivity (see Table 1 below for a summary of drug doses and effects in animal models). In 2018, Lovett-Racke and colleagues demonstrated that 20 mg/kg of CBD administered intraperitoneally (IP) significantly reduced the amounts of pro-inflammatory infiltration of interferon γ (IFN-γ) and interleukin-17 (IL-17) in female C57BL/6 mice who have been administered MOG (35–55). They found that treatment with CBD leads to the induction of myelin derived suppressor cells and the suppression of MOG specific T cell proliferation. Similarly, by depleting myelin derived suppressor cell levels, the beneficial effects of CBD were completely reversed.

Gonzáles-García and colleagues used an imaging approach to assess the benefits of CBD in EAE infected C57BL/6 mice. They found a reduced apparent diffusion coefficient in the basal ganglia, hippocampus, and the corpus callosum in all CBD-treated mice, but most notably in the higher dose group which was treated with 50 mg/kg. From the histological analysis, it was found that CBD alone significantly reduced axonal damage, cell infiltration and demyelination while also decreasing the levels of IL-6, a pro-inflammatory cytokine. Reduced microglial activity was also observed. Results showed that CBD caused decreased amounts of upregulation of ionised calcium-binding adaptor molecule 1 (IBA-1) labelling in adoptive transfer EAE mice and CB_2_ receptors were expressed in high levels in the microglia suggesting that the upregulation of CB_2_ receptors is important in MS. Similarly, Grassi et al. have demonstrated that CBD is anti-apoptotic against the neurodegenerative aspects of MS [66]. These authors used the clinical disease score (1–6) to measure the beneficial effects of CBD and found that EAE mice treated with CBD showed a greater trend of recovery compared to the EAE only mice. They also found that by treating mice with CBD, they were able to preserve the paw sensibility response in EAE mice. Histological analysis from this study revealed that mitogen-activated protein kinases (MAPK) signaling pathway is strongly activated by EAE induction, and, repeated administration of CBD reduces the levels of expression and thwarts neuronal death. This is consistent with the findings of Fitzpatrick and Downer who have written extensively on the topic [67].

Similarly to findings from the studies presented above, Nichols and colleagues [58] found that treating EAE induced mice with oral CBD earlier than other studies reduced the clinical disease score, again in C57BL/6 mice. Histological analysis from this study showed that mice which had received treatment with oral CBD had significantly reduced levels of myelin derived suppressor cells. Similar to these articles, it was reported that there were fewer cell infiltrates and T cells in the white matter tracts of the cerebellum. From this, the authors concluded that CBD has a role in reducing neuroinflammation in the cerebellum. This study reported an increase in IFN-γ which conflicts with the findings of Lovett-Racke [56] who reported that CBD can reduce the amounts of IFN-γ available. 

Kozela et al. differed in their approach in that they decided on a low dose of 5 mg/kg of CBD from studies on rheumatic arthritis that use cannabinoids. Even with a low dose of CBD, their histological analysis showed diminished axonal damage, as well as microglial activation and T-cell activation in the spinal cord with reductions in clinical scores being visible from day 23 post EAE induction. Rahimi et al. found very similar results and found that treating EAE-infected mice with CBD caused a reduction in neurobehavioural scores, reduced inflammation, demyelination, axonal damage, and cytokine expression in C57BL/6 mice treated with a low dose of 5 mg/kg administered IP. The authors found that these effects were visible on day 16 and concluded that CBD prevents against oxidative neuronal damage and neurodegeneration. These studies show that CBD is in fact effective in delaying symptom onset of EAE and slows disease progression.

### 2.3. Combinations of Δ^9^-THC and CBD Are More Effective in Symptom Relief and Neuroinflammation Reduction

From reviewing the literature, it appears that, a combination of Δ^9^-THC and CBD is more effective in reducing the symptoms associated with an animal model of MS compared to administration of Δ^9^-THC or CBD *per se*. This formulation has been achieved in the drug (6a*R*,10a*R*)-6,6,9-trimethyl-3-pentyl-6a,7,8,10a-tetrahydrobenzo[c]chromen-1-ol;2-[(6*R*)-3-methyl-6-prop-1-en-2-ylcyclohex-2-en-1-yl]-5-pentylbenzene-1,3-diol (nabiximols; Sativex). Sativex is a herbal preparation containing a defined quantity and standardized extracts of Δ^9^-THC and CBD formulated for oromucosal spray administration with potential analgesic activity, together with other minor cannabinoids, flavonoids, and terpenes, all derived from two cannabis plant varieties. In a 2012 study, researchers compared the effects of Sativex, a compound combining Δ^9^-THC and CBD in a 1:1 ratio, to a positive control of baclofen, a frontline treatment for muscle spasticity associated with MS or spinal cord injury [61]. Five mg/kg of Sativex were administered SC with the benefits being seen within 10 min and lasting for at least two hours. It was reported that this administration reduced peak spasticity by 20%. In another group, 10 mg/kg of Sativex were also administered SC. In this group, a peak reduction in spasticity of 40% was observed. The benefits observed in the higher dose group were the same as was observed in the baclofen positive control group, where a 40% peak spasticity reduction was found. It was deduced that Δ^9^-THC caused this reduction of spasticity due to its CB_1_ agonistic properties whereas CBD reduced the levels of cell infiltrates due to its PPAR-gamma association. The authors reported mild sedation as the only side effect of cannabis treatment. This is a great advantage over the current frontline treatment of baclofen which has several notable side effects such as seizures, drowsiness, fatigue, abdominal pain, underdose and overdose [30,31]. If this self-titrating oromucosal drug, which allows the user to achieve their own optimum dose, is as effective as current frontline treatments such as baclofen, but exhibits more favourable side-effects, does this drug have potential to be the new first-line treatment?

More recently in 2015, Moreno-Martet and colleagues [62] administered Sativex at 10 mg/kg. The authors reported that Sativex significantly mitigated the neurological symptoms in EAE, while reducing the number and extent of cell aggregates in the spinal cord of these mice. It was found that CBD alone would only delay symptom onset and did not reduce cell aggregates. This conflicts with the literature depicted previously, where it was found that CBD does indeed reduce cell aggregation infiltrating the spinal cord.

In 2019, several authors began to examine in-depth, the effects of administering Δ^9^-THC and CBD in mice. As seen in other studies, Al-Ghezi et al. [63] administered a 10 mg/kg combination of Δ^9^-THC and CBD in a 1:1 ratio. Their results showed that the combination of the two compounds reduced neuroinflammation and suppressed cell infiltration of TH1 and TH17 pro-inflammatory cytokines, as well as reducing IL-17, IFN-γ, Tumour necrosis factor α (TNF-α) and IL-Iβ while suppressing pro-inflammatory phenotypes. This is somewhat consistent with the literature described above which found that CBD alone could reduce the amounts of pro-inflammatory cytokines and promote anti-inflammatory cytokines. Interestingly, the authors found that 20 mg/kg of Δ^9^-THC or CBD only delayed the onset symptoms and only CBD or Δ^9^-THC, or Δ^9^-THC alone could attenuate neurological disability. Supporting the findings of Moreno-Martet et al., Al-Ghezi and colleagues found that treating CB_1_- and CB_2_-deficient mice with Δ^9^-THC or CBD had no beneficial effect. In another 2019 publication by Al-Ghezi and colleagues studying the effects of cannabinoids on the gut microbiome, it was reported that a combination of Δ^9^-THC and CBD caused reductions in pro-inflammatory cytokines IL-17 and IFN-γ, and mucin degrading bacterial species, while promoting anti-inflammatory IL-10 and transforming growth factor β (TGFβ). The authors reported that cannabinoids suppress neuroinflammation in an animal model of MS by preventing the microbial dysbiosis seen in EAE [63]. In this study, it was also found that the combination drug increased the amount of myelin derived suppressor cells in the periphery of the gut microbiome, most notably in the spleen.

Furthermore, in 2019, Zhou and colleagues studied the effects of combining Δ^9^-THC and CBD [65]. They found that these compounds significantly reduced tail withdrawal times from a painful stimulus. The authors found that cannabinoid compounds caused significant improvements in neurological disability scores and behavioural scores by reducing TNF-α and increasing brain-derived neurotrophic factor (BDNF). This article conflicts with the findings of the articles reviewed in this section as the authors did not report a statistically significant difference, but a marked difference, in the reductions of left and right paw withdrawal from a painful stimulus between the Δ^9^-THC and CBD group and the CBD group.

It is clear from the literature that Δ^9^-THC alone and Δ^9^-THC and CBD combinations are more effective in reducing disability and spasticity associated with animal models of MS. Higher doses of the combination drugs Δ^9^-THC and CBD, with 10mg/kg of each compound or higher (e.g., 20 mg//kg) proved to be most effective. These are doses that are well within the range of tolerance in humans.

## 3. Human Studies

### 3.1. Cannabinoids Are Only Moderately Effective in Reducing MS Related Spasticity

Hyperactive spinal reflexes cause velocity-dependent tonic stretch reflexes with significant tendon jerks, (more commonly known as spasticity) is a significant cause of disability in pwMS [35,36]. The literature shows that cannabinoids can reduce spasticity in EAE models of multiple sclerosis in rats, but the literature only partly supports the use of cannabinoids in reducing symptoms of spasticity in humans.

In support of the findings of animal studies, Marinelli, Mori, Canneva et al. examined the effects of a Δ^9^-THC:CBD spray in pwMS at baseline and after a 4-week treatment phase. After the 4-week treatment phase, a significant reduction in electromyography data was reported in 20 out of 36 patients (56%). No significant difference was reported in 22% and a significant increase was reported in the other 22% of patients. The authors reported a significant reduction in the muscle stretch reflex and a significant reduction in the Modified Ashworth scale (MAS) during treatment in Δ^9^-THC:CBD responders. With regard to this study, it should be noted that patients were still actively receiving treatment with first-line spasticity medications (mainly baclofen), which may have influenced subjective and objective spasticity results.

Further supporting the findings of animal studies, Markovà and colleagues studied the effects of Sativex as an add-on therapy in pwMS in a two-phase study design [68]. Participants were permitted to up-titrate their dose to a maximum of 12 sprays a day until optimal symptom relief was observed. Patients who reported ≥20% improvement in spasticity numerical rating scale (NRS) were classed as responders in phase A (70.5% of the sample) and were randomised into either placebo or treatment groups in phase B. When compared to baseline measures, these responders had significantly improved mean spasticity NRS, mean pain NRS and MAS scores when compared to patients receiving placebo. The authors concluded that in patients with moderate to severe resistant MS spasticity who initially responded to Δ^9^-THC:CBD spray during the 4 week trial period, adding Δ^9^-THC:CBD spray to already optimised antispasticity treatment is a better alternative to simply readjusting first-line medications only.

Similar to the design of the study described above, Ferrè et al. also implemented a two-phase design; a four-week titration phase followed by a treatment phase. Again, pwMS who had ≥20% reduction in spasticity NRS in the four-week phase A were classed as responders and were continued to phase B. The authors reported that 90.6% of patients classified as responders to Sativex in phase A, continued to show clinical benefit and good tolerability to the drug treatment even beyond the 14 weeks follow up period. These benefits were still evident at a one year follow up on spasticity NRS scores and MAS scores, but not on tests of walking ability, most likely due to its low sensitivity. These results suggest that Nabiximols are as effective as first-line treatments for spasticity.

Koehler et al. [69] studied the effects of the Sativex (i.e., Δ^9^-THC:CBD) spray as either an add-on therapy in 95 patients or as a monotherapy in 25 patients who had experienced weakness or instability on first-line anti-spasticity medications such as tizanidine or baclofen. The authors reported a mean reduction of 57% in the NRS (mean scores of 7.0 to 3.0) within 10 days of the therapy commencing. It was shown that patients receiving Δ^9^-THC:CBD as a monotherapy did not develop an increase in muscle weakness and showed NRS score reductions similar to those receiving the drug as an add-on therapy. It should be noted that 23 patients (13.9%) withdrew from the study due to adverse reactions such as dizziness, vertigo, fatigue, and incontinence. Despite these adverse reactions, the authors still reported that the drug combination was well tolerated (see Table 2 below for a description of the adverse events reported by authors).

These results were only partially replicated by Leocarni and colleagues. When compared with patients who received a placebo, significant improvements were found on the MAS which is generally regarded as less sensitive to changes in spasticity. Despite this positive change, in measures of neurophysiology, no significant difference in change from baseline was reported in patients receiving either placebo or Sativex [70]. Similarly, no significant difference in change in objective NRS scores was reported.

Interestingly, Van Amerongen and colleagues studied the effects of cannabinoids on MS-related spasticity slightly differently than the studies listed above by using ECP002A, a synthetic Δ^9^-THC drug [71] (see Table 3 for a list of cannabinoid ligands used in human studies). In contrast to the animal studies, these authors reported that there was no significant difference in muscle reflex scores, spasticity, or pain NRS scores, or MAS scores between patients receiving ECP002A and patients receiving placebo. Slightly worryingly, 200 adverse events were recorded by the authors, most have been classified as mild, nine treatment-emergent adverse events were classified as moderate and one diagnosis of euphoric mood was made and classified as severe as it significantly disrupted the patient’s day to day living and work. These mild adverse events consisted mostly of dizziness; euphoric mood (high feeling), headache, somnolence, and fatigue.

To conclude this subsection, the evidence only partially supports the notion that cannabinoid compounds such as those which deliver Δ^9^-THC and CBD in a 1:1 ratio, or drug formulations which deliver Δ^9^-THC alone reduce MS-related spasticity.

The polyphenolic nature of CBD makes it a potent antioxidant. Thus, though the Sativex combination, CBD may potentiate some of the beneficial effects of Δ^9^-THC, as it reduces the psychoactivity of Δ^9^-THC to enhance its tolerability and widen its therapeutic window. CBD can counteract some of the functional consequences of CB_1_ activation in the brain, possibly by indirect enhancement of adenosine A1 receptors activity through ENT inhibition. This effect could partly explain why users of cannabis preparations with high Δ^9^-THC and CBD ratios are less likely to develop psychotic symptoms than those who consume preparations with low Δ^9^-THC and CBD ratios. Sativex relieves spasticity and pain in multiple sclerosis more effectively than Δ^9^-THC alone, possibly because CBD’s effects allow patients to tolerate higher amounts of Δ^9^-THC. CBD may also supplement the antispastic effects of Δ^9^-THC, possibly through local potentiation of glycine signaling, inhibition of endocannabinoid degradation, or retardation of demyelination through anti-inflammatory, antioxidant, and anti-excitotoxic mechanisms [79].

### 3.2. Cannabinoids May Be Effective in Reducing Neuropathic Pain in pwMS

Neuropathic pain (NP) is thought to be caused by a lesion in, or dysfunction of the CNS, possibly as a result of corticospinal system disinhibition or chronic activation of nociceptive afferents, and is a common symptom of MS affecting between 17% and 70% of patients [47,48].

In a study by Furri et al., 28 pwMS were recruited for a two-phase study [73]. These patients were administered Sativex for one month. This study aimed to examine the possibility of Sativex reducing pain, which was quantified by NRS scores, laser evoked potentials were also used to detect subcortical lesions. Seventy four percent of patients responded to Sativex at the end of phase A, but there was no significant difference between pain NRS scores and laser evoked potential amplitude between Sativex treatment group and baseline scores. The results of phase B showed that Sativex was effective in relieving pain NRS scores and caused a reduction in amplitude in laser evoked potentials. The results of this study suggest that Sativex is an effective method of reducing NP in pwMS, despite this, it should be noted that this study did not implement a placebo group and the observation period was relatively short.

A more favourable study design was implemented by Langford et al., who implemented a double-blind placebo-controlled parallel-group design in a large sample size of 393 patients [74]. In phase A of this study, the double-blind placebo group, patients were given either Sativex or a control substance. Each 100 µL activation of the spray delivered 2.7 mg of Δ^9^-THC and 2.5 mg of CBD to the oral mucosa. Patients were restricted to a maximum of 12 sprays a day. In phase A, those receiving Sativex self-administered a mean of 8.8 sprays a day compared to a mean of 11.1 in the placebo group. Fifty percent of patients in the Sativex group displayed a 30% reduction in pain NRS scores, whereas 45% of patients in the control group displayed a similar reduction. At the end of this phase, patients in the treatment group had a mean pain NRS reduction of 1.93 points, and patients in the control group had a mean reduction of 1.76 points. Similarly, none of the secondary endpoints of this study (Brief Pain Inventory - Short Form, Subject Global Impression of Change) and sleep quality assessments displayed a statistically significant difference in favour of the treatment. These results did not reach statistical significance, but, were clinically relevant. At the end of phase B of this study, the open-label phase, only 24% of patients failed treatment with Sativex compared to 57% in placebo, suggesting the efficacy of primary endpoints (NRS scores) and some secondary endpoints.

In another study, Russo et al. also studied the effects of Sativex in 10 pwMS with NP and 10 pwMS without NP [79]. These researchers evaluated the drug effects on a pain NRS, MAS and by neurophysiological measures. During treatment, all patients were taking anti-convulsant drugs (most commonly baclofen), 40% of these patients occasionally took analgesics for pain relief. After one month of drug administration, pwMS with NP showed significant reductions in pain NRS scores and showed improved scores on measures of quality of life. These effects were salient with the neurophysiological findings which showed an increase of frontocentral γ-band oscillation and pain motor integration strength. The data from this study supports the effectiveness of Sativex in pain relief in NP pwMS and improved quality of life. These authors suggest that Sativex may also boost cortical pain gating mechanisms as clinical pain relief was salient with the findings of activation of sensory-motor areas supporting the idea that Sativex may restore cortical pain gating mechanisms, most likely through modulation of sensory-motor integration concerning painful stimulation [37,55].

A 2005 study by Rog et al. studied the effects of nabiximols on central NP in a randomised control trial [80]. Sixty-six patients were randomised to either Sativex or control groups and asked to complete pain NRS daily from baseline to end of treatment. No specific target dose was set, and patients were advised to self-titrate the number of sprays in a stepwise manner on consecutive days to a maximum of 48 sprays a day. This is the highest nabiximols dose administered in any study reviewed in this manuscript. Individuals in the Sativex treatment group self-administered a mean 9.6 sprays a day compared to 19.1 sprays in the control group. The authors reported that Sativex caused a significant reduction in pain NRS scores suggesting the efficacy of the drug. Conversely, no statistically significant difference, but a clinically relevant difference was found between treatment groups and placebo groups were reported in secondary outcomes (hospital anxiety depression scale, Guy’s neurological disability scale). Remarkably similar results were found by Schimrigk et al., who studied the effects of dronabinol on NP in pwMS [75]. Here dronabinol dosing was increased every five days by 2.5 mg to reach a daily dose between 7.5 mg to 15 mg (mean 12.7 ± 2.9 mg). Similar to above, the authors found a clinically relevant difference between placebo and treatment groups without reaching statistical significance. The authors concluded that dronabinol is a safe long-term treatment option for NP due to its analgesic, sedative, spasmolytic, anti-inflammatory, and anxiolytic effects, all of which contribute to improved scores on quality of life measures.

### 3.3. Cognitive Effects of Cannabinoid Treatment

Cognitive dysfunction can affect up to 80% of pwMS and can have severe knock-on effects such as difficulty maintaining employment, sustaining a meaningful relationship, as well as significantly affecting the individual’s quality of life [81], does cannabis exacerbate this dysfunction?

Not only have Amerongen et al. used cannabinoids to examine spasticity in pwMS [71], they have also used them to examine the possible cognitive implications of cannabinoid treatments in this clinical population. They reported that treatment with cannabinoids did not cause a statistically significant decline of postural stability, cognitive functioning, mood or psychomimetic effects. Despite this, they did find a slightly clinically relevant deterioration in attention and cognitive functioning had occurred during treatment with ECP002A. These findings are similar to those reported by Massimiliano et al. [76] who reported that their sample of 17 pwMS receiving Sativex (8.2 mean sprays a day) did not score significantly differently on measures of cognitive performance [Paced Auditory Serial Addition Test (PASAT); Multiple Sclerosis Functional Test (MSFT)] or quality of life measures [Visual Analogue Scale (VAS); fatigue severity scale, multiple sclerosis impact scale] in either the placebo or treatment phases. Despite not finding statistical significance in objective measures, a clinically relevant impairment was observed where 11 patients receiving Sativex reported subjective drowsiness and a sense of slower thinking.

These non-significant results conflict with the findings of Honormand et al. [82] who reported that cannabis users scored significantly lower on PASAT, judgement of line orientation, symbol digit modalities test and the Delis-Kaplan Executive Function System (D-KEFS) when compared to placebo, and on tests of premorbid functioning in a sample of 50 pwMS (25 smoked cannabis users, 25 receiving placebo). Patients were asked to abstain from cannabis use before testing. Patients abstained from use for between 12 h to several days. Considering the half-life of Δ^9^-THC is between 5 and 13 days for regular cannabis users [83] and metabolites being present in urine samples for up to 46 days [57], this begs the question; is this abstinence period long enough to not affect the results? The authors reported that cannabis users were twice as likely to meet the criteria for global impairment, which the authors defined as failure on two of the eleven tests used, as well as being twice as likely to be unemployed. Similar results were reported in a 2017 study by Patel and Feinstein who examined the cognitive functioning in 140 pwMS who smoke cannabis monthly or more frequently who found that cannabis users had significantly worse on the symbol digit modalities test and on the total learning and long-delayed free recall indices of the California learning test-II [84].

Feinstein et al. used the Brief Repeatable Neuropsychological Battery (BRNB) for MS along with fMRI data to assess for cognitive dysfunction in 40 patients who began using cannabis after they received their diagnosis, all of whom report using cannabis on average, four times a week for NP, spasticity, incontinence, insomnia and recreational use. Participants were only included in this study if they were deemed by the authors to be globally cognitively impaired, which was classified as failure on two or more domains of the BRNB. Patients were assigned to either cannabis abstinence or cannabis continuation groups and underwent neuropsychological assessment and brain imaging at baseline and after 28 days. The authors reported that by the end of the study period, patients in the abstinence group showed significantly better scores on every cognitive index of the BRNB. Furthermore, the cannabis abstinence group reported more withdrawal symptoms and fewer depressive symptoms, but these changes did not reach statistical significance. These findings were supported by fMRI data where higher activations were found in the right inferior frontal gyrus, left inferior frontal gyrus, right declive cerebellum and right caudate nucleus in the abstinence group. The abstinence group also revealed significantly increased blood oxygen level-dependent related activations in the right middle frontal gyrus, left inferior frontal gyrus, right precuneus and left ceneus. The authors of this study concluded that cannabis use causes significant cognitive dysfunction in pwMS and a short period of abstinence was associated with better performance on tests of information processing speed, executive functioning, learning and memory. These improvements were matched by increases in cerebral activation on fMRI.

These fMRI findings were supported by a study published by Pavisian et al. [85] who also administered the BRNB for MS in 20 cannabis-using patients (asked to refrain from using the drug for at least 12 h prior to testing) and 19 cannabis naïve patients. Similar to the above, patients were deemed to be globally impaired if they failed two or more indices of the BRNB. The N-back test was used during fMRI and subjects were asked to respond with a two-button response pad. The authors reported that there was no significant difference in 0- and 1-back trials but on 2-back trials, it was found that cannabis users achieved fewer correct responses, but no significant difference in reaction times. Further, cannabis users scored significantly lower on the BRNB showing that users have more cognitive deficits than those who do not.

Significant cognitive deficits in pwMS who smoke cannabis were also found by Romero et al. [86]. They reported that cannabis users achieved significantly lower scores on tests of spatial memory and significantly lower scores of information processing speed. Moreover, the significantly poorer neuropsychological test scores, fMRI data showed that grey and white matter volume in the medial and temporal regions, thalamus, basal ganglia and prefrontal cortex was associated with significantly more cognitive deficits in the 20 pwMS who smoke cannabis regularly compared to the 19 pwMS who do not. It was also reported by these authors that decreased brain volume was correlated with poorer performance on all neuropsychological tests in cannabis-smoking pwMS compared to non-cannabis smoking pwMS.

## 4. Conclusions

From the literature discussed, it is accurate to say that CBD is highly effective in slowing the onset of symptoms and the progression of disease in EAE mice that have been induced with MOG (35–55). It seems to do this by reducing the amounts of pro-inflammatory cells, microglial and T-cell activation as well as diminished axonal damage in the spinal cord and reduced cytokine expression. Additionally, it seems that when CBD is administered IP, it protects the mouse from oxidative neuronal damage and neurodegeneration. It would be inaccurate to draw from the literature which dose is most effective in doing this, as conflicting findings have repeatedly been reported, with some studies having reported doses as low as 5 mg/kg showing effective and promising results [56], whereas others reporting that only higher doses, up to 50 mg/kg, were effective [57]. Building from this, the literature clearly shows that administering drug combinations of CBD and Δ^9^-THC, such as nabiximols (Sativex), are more effective in reducing the symptoms of MS in EAE-induced mice than administering either CBD or Δ^9^-THC alone. This may be due in part to the compounds having a synergistic effect. While CBD reduces the amount of infiltration of pro-inflammatory cytokines: IL-17, IFN-γ, TNF-α and IL-Iβ [87], Δ^9^-THC seems more effective in reducing clinical scores of spasticity.

The results of human studies are far less encouraging. A drawback for this cohort of studies is the fact that histological analysis of cell aggregates in the spinal cord cannot be done in living humans. The literature in this area shows that cannabinoid drugs such as Sativex or ECP002A are moderately effective in reducing scores of spasticity as measured by spasticity NRS or MAS. This literature is conflicting and often difficult to decipher. A possible cause of the conflicting nature of these studies is the study design. Rightly, some studies implement a randomised double-blind placebo-controlled study design in a large sample size [88,89], whereas others implement less favourable study designs in smaller groups. These varying designs pose significant issues for prescribing physicians as the results pose uncertainty. A slightly clearer picture is painted regarding NP. Sativex seems to be effective in reducing patient-reported measures of pain, as measured by pain NRS, but less effective in reducing objective measures of pain, measured by the MAS. Again, these findings can be contradictory. Some studies report statistically significant results whereas others report clinically relevant or non-significant results. Contradictory findings can also be seen in terms of cognition. Again, some authors report statistically significant cognitive dysfunction and cortical changes as a result of Sativex or smoked cannabis, whereas others found that cannabis use causes little to no cognitive or cortical changes.

Further research is needed in this field, as currently, the findings are about as clear as mud. In animal studies, an optimal dose should be found to assess the true effect of CBD and Δ^9^-THC on models of MS. In humans, more randomised double-blind placebo-controlled studies are needed in a long-term patient group. By doing this, researchers will be provided with more in-depth knowledge of the effects of cannabinoid drugs on spasticity, pain, and cognition.

## Figures and Tables

**Figure 1 molecules-25-04930-f001:**
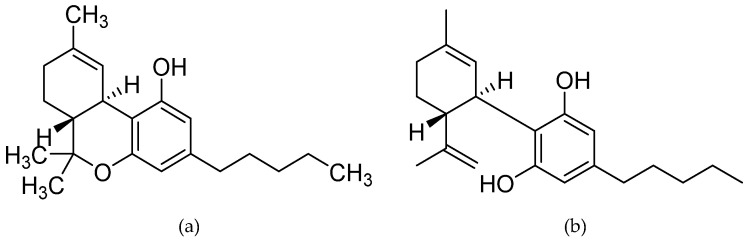
Chemical structure of the two most studied cannabinoid molecules; (**a**) Δ^9^-tetrahydrocannabinol, and (**b**) cannabidiol.

**Table 1 molecules-25-04930-t001:** Summary of the effects of cannabinoids on EAE mouse models of multiple sclerosis.

Ligand (Route)	Concentration (Time of Administration in Days Post Disease Induction)	Species (Sex)	Effect	Reference
CBD (IP)	20 mg/kg (9–25)	C57BL/6 (f)	↑	[56]
CBD (IP)	5–10 mg/kg (0)	C57BL/6 (f)	↑	[57]
	10 mg/kg (0)			
	50 mg/kg (0)			
CBD (OG)	75 mg/kg (1)	C57BL/6 (f)	↑	[58]
CBD (IP)	5 mg/kg (19–21)	C57BL/6 (f)	↑	[59]
CBD (IP)	5 mg/kg (11–13)	C57BL/6 (f)	↑	[60]
CBD:Δ^9^-THC (IV)	5 mg/kg (210)	Biozzi ABH (m/f)	↑	[61]
	10 mg/kg (210)			
CBD:Δ^9^-THC (SC)	10 mg/kg (11)	C57BL/6 (f)	↑	[62]
Δ^9^-THC (SC)	20 mg/kg (11)	C57BL/6 (f)	↑	[62]
CBD (SC)	20 mg/kg (11)	C57BL/6 (f)	—	[62]
CBD:Δ^9^-THC (SC)	10 mg/kg (10–15)	C57BL/6 (f)	↑	[63]
CBD (SC)	20 mg/kg (10–15)	C57BL/6 (f)	—	[63]
Δ^9^-THC (SC)	20 mg/kg (10–15)	C57BL/6 (f)	↑	[63]
CBD:Δ^9^-THC (IP)	10 mg/kg (10–15)	C57BL/6 (f)	↑	[64]
CBD:Δ^9^-THC oil extract (OG)	215 mg/kg (6–18)	Lewis (f)	↑	[65]
CBD oil extract (OG)	215 mg/kg (6–18)	Lewis (f)	—	[65]
Δ^9^-THC oil extract (OG)	215 mg/kg (6–18)	Lewis (f)	—	[65]

IP = intra peritoneal; OG = oral gavage; IV intravenous; SC = subcutaneous; f = female; m = male; ↑ = positive reduction in symptomatology and histological markers; — = no significant change in symptomatology and histological markers.

**Table 2 molecules-25-04930-t002:** Frequencies of adverse events of MS patients treated with cannabinoids from studies where frequencies were reported.

Adverse Event	[69]	[70]	[71]	[72]	[73]	[74]	[75]	[76]	Total
Dizziness	5	7	13	35	4	34	50	4	152
Headache	0	0	9	0	0	7	9	3	208
Somnolence	0	0	9	11	0	16	0	0	36
Muscle Weakness	3	2	5	0	0	1	0	3	14
Spasticity	0	0	3	4	0	0	0	0	7
Paraesthesia	0	0	2	0	0	0	0	0	2
Tremor	0	0	3	0	0	0	0	1	4
Vertigo	2	1	0	0	0	16	34	4	57
Tinnitus	0	0	2	0	0	0	0	0	2
Mood Disruption	0	0	0	0	0	2	0	1	3
Euphoria	0	0	9	0	0	0	0	1	10
Attention	0	0	1	0	2	6	0	11	20
Insomnia	0	0	1	0	0	0	2	0	3
Fatigue	5	0	5	29	2	16	25	6	88
Feeling abnormal	0	0	5	50	0	5	0	0	60
Feeling hot	0	0	3	0	0	0	0	0	3
Oral Discomfort	4	0	3	0	0	19	13	5	44
Nausea	2	0	1	0	0	12	17	2	34
Appetite	0	0	2	3	0	0	0	0	5
Stomatitis	1	0	0	0	0	0	0	0	1
Incontinence	1	0	0	0	0	0	0	0	1
Hypertension	0	1	0	0	0	8	0	0	9
Pharyngodynia	0	1	0	0	0	2	0	0	3
Vision Blurred	0	0	0	0	0	4	0	0	4
Diarrhoea	0	0	0	0	0	7	13	0	20
Vomiting	0	0	0	0	0	5	0	0	5
Memory Impairment	0	0	0	0	0	6	0	0	6
Psychomotor Impairment	0	0	0	0	0	5	0	0	5
Total Adverse Events									806
Total Participants	166	22	24	144	28	312	333	17	

**Table 3 molecules-25-04930-t003:** Cannabinoid ligands used in human studies of spasticity, pain and cognition.

Ligand	Chemical Name	Reference
Nabiximols (Sativex)	(6a*R*,10a*R*)-6,6,9-trimethyl-3-pentyl-6a,7,8,10a-tetrahydrobenzo[c]chromen-1-ol;2-[(6*R*)-3-methyl-6-prop-1-en-2-ylcyclohex-2-en-1-yl]-5-pentylbenzene-1,3-diol	[77]
Δ9-tetrahydrocannabinol (Dronabinol)	(−)-(6a*R*,10a*R*)-6,6,9-trimethyl-3-pentyl-6a,7,8,10a-tetrahydro-6H-benzo[c]chromen-1-ol	[78]
ECP002A	(−)-(6a*R*,10a*R*)-6,6,9-trimethyl-3-pentyl-6a,7,8,10a-tetrahydro-6H-benzo[c]chromen-1-ol	[77]

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
