# Peer review of "A Critical Review of the Role of the Cannabinoid Compounds Δ^9^-Tetrahydrocannabinol (Δ^9^-THC) and Cannabidiol (CBD) and their Combination in Multiple Sclerosis Treatment"

_molecules, 2020, doi:10.3390/molecules25214930_

Round 1
Reviewer 1 Report
This is a well-written manuscript/review and addresses key components well, except a couple of issues outlined below.
- Line 21: Should it be "α2 adrenergic receptor agonist", rather than "a2 adrenergic receptor agonist"?
- Section 2. Animal Studies. Overall, this section is written well, but some key models are missing and authors must include comments for full coverage. There are neurodegenerative models such as cuprizone model, and cuprizone autoimmune encephalomyelitis (CAE) model, which are relatively recent and mimic the progressive phase features of the MS disease. I would recommend authors include either a sub-section, or a rational description of these models, and relevance to cannabinoid treatment, if any. Just for authors' information: (a) Kipp M, Clarner T, Dang J, Copray S, Beyer C. The cuprizone animal model: new insights into an old story. Acta Neuropathol. 2009;118(6):723-736. doi:10.1007/s00401-009-0591-3 and (b) Caprariello AV, Rogers JA, Morgan ML, et al. Biochemically altered myelin triggers autoimmune demyelination. Proc Natl Acad Sci U S A. 2018;115(21):5528-5533. doi:10.1073/pnas.1721115115.
Author Response
Reviewer 1
- “This is a well-written manuscript/review and addresses key components well, except a couple of issues outlined below.”
We would like to thank the reviewer for this positive comment on our effort.
- “Line 21: Should it be "α2 adrenergic receptor agonist", rather than "a2 adrenergic receptor agonist"?”
Definitely yes. We appreciate this attention to detail and identification of this mistake at this stage. The wording has now been changed to "α2 adrenergic receptor agonist". Please see this change reflected on p.1, Abstract, line 21.
- “Section 2. Animal Studies. Overall, this section is written well, but some key models are missing and authors must include comments for full coverage. There are neurodegenerative models such as cuprizone model, and cuprizone autoimmune encephalomyelitis (CAE) model, which are relatively recent and mimic the progressive phase features of the MS disease. I would recommend authors include either a sub-section, or a rational description of these models, and relevance to cannabinoid treatment, if any. Just for authors' information: (a) Kipp M, Clarner T, Dang J, Copray S, Beyer C. The cuprizone animal model: new insights into an old story. Acta Neuropathol. 2009;118(6):723-736. doi:10.1007/s00401-009-0591-3 and (b) Caprariello AV, Rogers JA, Morgan ML, et al. Biochemically altered myelin triggers autoimmune demyelination. Proc Natl Acad Sci U S A. 2018;115(21):5528-5533. doi:10.1073/pnas.1721115115.”
We would like to thank the reviewer for pointing out this important omission. We have now included information on the CAE model and how it relates to other models as a subsection of the 2.1.2 section “Toxin-Induced Demyelination”. We have also supported this information with relevant literature, including the two published articles that the reviewer suggested. Please see reflected changes on p.4-5, lines 176-199.
Reviewer 2 Report
In general, the manuscript is well done and structured and summarizes recent findings in animal model and human trials for the treatment of MS. However, due to the aims of the journal (the leading international peer-reviewed open access journal of chemistry) the "chemistry" of the selected compounds is completely missing, but mandatory in this context.
In my opinion, without this part the manuscript is more suitable to a medicine journal (example INT J MOL SCI.) than to a chemistry journal.
MINOR POINT
Sativex is a drug with 1delta9-THC:1CBD ratio. In this respect, sentences like "Cannabinoids such as Sativex" are not appropriate within the manuscript.
I suggest the use of CB1 for cannabinoid 1 receptor (and not receptors)and CB2 for cannabinoid 2 receptor (and not receptors).
In spite the definition of CB1 and CB2 abbreviation, sometimes the word "receptor" still follows the abbreviation.
Table 2: In the title specify MS patients
Table 3: In the title specify the matter of the study
In review articles, inclusion of figures is recommended.
Author Response
Reviewer 2
- “In general, the manuscript is well done and structured and summarizes recent findings in animal model and human trials for the treatment of MS. However, due to the aims of the journal (the leading international peer-reviewed open access journal of chemistry) the "chemistry" of the selected compounds is completely missing, but mandatory in this context. In my opinion, without this part the manuscript is more suitable to a medicine journal (example INT J MOL SCI.) than to a chemistry journal.”
We would like to thank the reviewer for their overall positive evaluation of our manuscript.
We very much appreciate the comment of the reviewer on the important omission of chemical aspects of this topic, seeing as it is submitted to the “Medicinal Chemistry” section of Molecules. We agree with the reviewer’s comment and, in this regard, we have made an effort to include more information on the chemistry of the two cannabinoid compounds extensively presented in this manuscript, Δ9-tetrahydrocannabinol and cannabidiol. Please see p.2 and p.3, lines 88-96, and p.3, lines 106-130, for these changes. Further, we have added a figure of the chemical structure of Δ9-tetrahydrocannabinol and cannabidiol (please see p.4, Figure 1) in the manuscript.
MINOR POINT
- “Sativex is a drug with 1delta9-THC:1CBD ratio. In this respect, sentences like "Cannabinoids such as Sativex" are not appropriate within the manuscript.”
We thank the reviewer for this comment. It was a mistake and it has now been corrected in the corresponding sections. Please see p.1, Abstract, lines 18-19, p.6, lines 270-276 and p.9, line 367 and lines 394-395.
- “I suggest the use of CB1 for cannabinoid 1 receptor (and not receptors)and CB2 for cannabinoid 2 receptor (and not receptors). In spite the definition of CB1 and CB2 abbreviation, sometimes the word "receptor" still follows the abbreviation.”
We thank the reviewer for this suggestion. We have now consistently refered to CB1 and CB2 for cannabinoid 1 receptor and cannabinoid 2 receptor, respectively, throughout the manuscript, and we hope we have not omitted any CB1 and CB2 references.
- “Table 2: In the title specify MS patients”
We thank the reviewer for this correction. “MS” has now been added before “patients” to specify which exact patient population this Table refers to.
- “Table 3: In the title specify the matter of the study”
We thank the reviewer for this comment. “Spasticity, pain and cognition” have now been added in the title for Table 3 to specify the focus of the studies.
- “In review articles, inclusion of figures is recommended.”
We agree with the reviewer on this comment and we appreciate this recommendation. However, although during our detailed literature search we identified a number of Figures focusing on animal models for MS, more specifically, informing the reader of the process of autoimmune demyelination, as found in experimental autoimmune encephalomyelitis and toxin induced demyelination with the cuprizone model (Denic et al., 2011, Figure 1; Caprariello et al., 2018, Figure 5), which could have been useful to this manuscript, we preferred to focus our efforts into creating Table 1 and Table 2 in the original submission. In this revised submission, we would have liked to include the Figures previously identified, either as originally published or as adapted, upoen permission, however, the short period of time to resubmit this manuscript (10 days) did not allow for requests for and receipt of permissions. We hope this manuscript can still be accepted without including other figures. Figure 1 has now been added, presenting the chemical structure of Δ9-tetrahydrocannabinol and cannabidiol.
Round 2
Reviewer 2 Report
The submitted review article summarizes the therapeutic application of Sativex, a synthetic combination of cannabidiol and Δ9 -tetrahydrocannabinol for the treatment of multiple sclerosis. The manuscript is well done and summarizes data from both animal models and humans in phisiological and pathological conditions.
The authors have included few information concerning the structure of both cannabidiol and Δ9 -tetrahydrocannabinol and have added a figure with the chemical structure of these compounds. Considering the large family of "cannabinoid compounds" in the title, this seem quite poor. Nevertheless, as for the previous version, the review seems better structured for a molecular/medicine journal and the current version is borderline for a chemistry journal. However, if the editor finds sufficient the inclusion of these few chemistry data, I have no trouble in endorsing the manuscript for publication.
In all cases, the title " A Critical Review of the Role of Cannabinoid Compounds in Multiple Sclerosis Treatment" need to be more focused on Sativex.
All the other minor queries have been addressed.
Author Response
Reviewer 2
- “The submitted review article summarizes the therapeutic application of Sativex, a synthetic combination of cannabidiol and Δ9 -tetrahydrocannabinol for the treatment of multiple sclerosis. The manuscript is well done and summarizes data from both animal models and humans in phisiological and pathological conditions.”
We would like to thank the reviewer for an overall positive evaluation of our manuscript.
- “The authors have included few information concerning the structure of both cannabidiol and Δ9 -tetrahydrocannabinol and have added a figure with the chemical structure of these compounds. Considering the large family of "cannabinoid compounds" in the title, this seem quite poor. Nevertheless, as for the previous version, the review seems better structured for a molecular/medicine journal and the current version is borderline for a chemistry journal. However, if the editor finds sufficient the inclusion of these few chemistry data, I have no trouble in endorsing the manuscript for publication.”
We respect the reviewer’s view on the content, and we appreciate that they now find it is borderline for a chemistry journal, seeing as it is submitted to the “Medicinal Chemistry” section of Molecules. We have made a further effort to include some more information on the chemistry of the two cannabinoid compounds extensively presented in this manuscript, Δ9-tetrahydrocannabinol and cannabidiol. Please see p.1 lines 1-4 (Title) p.3, lines 118-119 and 130-136, as well as p.6-7, lines 278-280, for these further minor changes.
- In all cases, the title "A Critical Review of the Role of Cannabinoid Compounds in Multiple Sclerosis Treatment" need to be more focused on Sativex.
The title has now been adjusted to “A Critical Review of the Role of the Cannabinoid Compounds Δ9-tetrahydrocannabinol (Δ9-THC) and Cannabidiol (CBD) and their Combination in Multiple Sclerosis Treatment”, focusing only on Δ9-THC, CBD and their combination treatment for MS. Please see p.1, lines 1-4.
- “All the other minor queries have been addressed.”
Thank you.